# Bacterial Proteases as Potentially Exploitable Modulators of SARS-CoV-2 Infection: Logic from the Literature, Informatics, and Inspiration from the Dog

**DOI:** 10.3390/biotech12040061

**Published:** 2023-10-30

**Authors:** Gerald H. Lushington, Annika Linde, Tonatiuh Melgarejo

**Affiliations:** 1Qnapsyn Biosciences, Inc., Lawrence, KS 66046, USA; glushington@qnapsyn.com; 2College of Veterinary Medicine, Western University of Health Sciences, Pomona, CA 91766, USA; alinde@westernu.edu

**Keywords:** TMPRSS2, protease–antiprotease balance, COVID-19, SARS-CoV-2, spike protein, *Moraxella*, *Pseudomonas*, commensal bacteria, serine protease, *Canis lupus familiaris*

## Abstract

(1) Background: The COVID-19 pandemic left many intriguing mysteries. Retrospective vulnerability trends tie as strongly to odd demographics as to exposure profiles, genetics, health, or prior medical history. This article documents the importance of nasal microbiome profiles in distinguishing infection rate trends among differentially affected subgroups. (2) Hypothesis: From a detailed literature survey, microbiome profiling experiments, bioinformatics, and molecular simulations, we propose that specific commensal bacterial species in the *Pseudomonadales* genus confer protection against SARS-CoV-2 infections by expressing proteases that may interfere with the proteolytic priming of the Spike protein. (3) Evidence: Various reports have found elevated *Moraxella* fractions in the nasal microbiomes of subpopulations with higher resistance to COVID-19 (e.g., adolescents, COVID-19-resistant children, people with strong dietary diversity, and omnivorous canines) and less abundant ones in vulnerable subsets (the elderly, people with narrower diets, carnivorous cats and foxes), along with bioinformatic evidence that *Moraxella* bacteria express proteases with notable homology to human TMPRSS2. Simulations suggest that these proteases may proteolyze the SARS-CoV-2 spike protein in a manner that interferes with TMPRSS2 priming.

## 1. Introduction

For all its rigor, science is also a discipline of intuition and adventition. Since the days of Friedrich Kekulé correctly perceiving the structure of benzene from his vision of a snake eating its tail, scientists have benefited from unexpected analogies that have suddenly emerged to explain odd observations. In an attempt to rationalize a disparate series of counter-intuitive epidemiological trends emanating from the COVID-19 pandemic, this manuscript develops a hypothesis that may help to resolve questions regarding idiosyncratic variations in human infectious vulnerability based, originally, on our observations on the varying microbiotic profiles of canines and felines.

Among the many ways in which COVID-19 has impacted humanity, the most profound one may actually prove beneficial—a period of prodigious scientific discovery aimed at advancing global health [1,2,3,4,5,6,7,8,9,10,11,12,13]. Adding to the disciplines influenced by pandemic-related research, this hypothesis explores intriguing evidence of microbe–virus interplay, combining verified epidemiological and biochemical reports with new preliminary analyses to argue for the role of microbial proteases in rationalizing coronavirus epidemiology trends, to illuminate potential opportunities for novel therapeutic strategies.

As the first pandemic of the information age and post-genomic era, COVID-19 brought tremendous investigative resources bearing on virology, immunology, and epidemiology, producing a stunning array of new insight that should help to spark new research for years to come. The greatest short-term technological impact of the COVID-19 response was likely the remarkable success of anti-SARS-CoV-2 vaccine programs, but other critical advances in viral testing and surveillance were also achieved with rapid effectiveness [1,2,3,4]. The protocols by which each of these high-impact COVID-19-specific technologies was attained were sophisticated and timely and offer excellent roadmaps for tackling future epidemics, thus bettering the future of world health.

Beyond the aforementioned developments, one must also consider advances in risk-profile analysis, producing important insight into why some subsets of the human population have demonstrated elevated risk of suffering more severe COVID-19 outcomes. Statistical support for many posited risk factors has declined through the pandemic, but some, such as a link between severe COVID-19 and prior medical history of cardiovascular and metabolic disorders (primarily diabetes, hypertension, and obesity) have gained strength according to extensive sustained analysis [5]. Autoimmune risk factors have also been determined to correlate with COVID-19 severity [6,7] and predispose patients to post-acute sequelae [8,9]. Other epidemiologically significant factors include chronic exposure to air pollution [10,11], as well as one metric that assimilates numerous subfactors—socioeconomic status [12,13].

Vaccination, testing, and risk profiling are of tremendous societal value and can substantially reduce overall caseloads and fatalities, but unfortunately, they offer marginal relief to individuals who actually contract dangerous viruses like COVID-19. Further development of safe and reliable treatment strategies for people who fall ill is thus required for full and concerted pandemic response. Much effort has been devoted to designing [14,15,16,17,18] or repurposing [19,20] drugs for SARS-CoV-2 targets, but such pursuits have not yet produced breakthrough success. Such disappointment is not surprising, given the historical challenges of developing antiviral therapeutics [21], but it is imperative to persevere and to innovate. The terrible global toll through a few short years of pandemic, the continuing threat of new coronavirus variants and hitherto undiscovered viral pathogens, and the rhetorical value of devising efficacious antiviral strategies all provide powerful incentives. The most pressing existential threats of COVID-19 may be on the wane, but our best chances to successfully mitigate the next pandemic may emerge from the vast amount of quantitative and qualitative evidence collected during the current one. Conceivably, this body of information may contain crucial mechanistic clues underlying past pharmacological failures and the basis for finding ways to correct them.

One superficially logical strategy for antiviral drug development that rarely tends to achieve lasting therapeutic benefits [22,23,24,25] entails applying standard rational small-molecule design principles toward interfering with viral proteins tasked with host infection or viral replication. One crucial limitation of this approach lies in ignoring long-standing experience with viral resistance, wherein high-specificity targeting is generally overcome by viral genetic evolution, via mutation rates substantially exceeding those of bacteria or cancer cells [26,27,28,29,30]. Alternative host-targeting strategies (e.g., blocking entry-mediating host–protein attachment sites) are less vulnerable to viral resistance, but the strategy has yet to produce major drug candidates [31,32]. Still, these outcomes should be regarded not as abject failures but rather as motivation to formulate new design paradigms. Every recent antiviral disappointment, such as Remdesivir [15], Sofosbuvir [16], and Molnupiravir [17], and controversies over hydroxychloroquine [19] and Ivermectin [20] add new insight into the merits and consequences of modulating viral–host interactions. Such incremental knowledge may eventually inform successful new strategies.

Valuable insights are also emerging from non-traditional hypotheses that may supplant or augment prior pharmacological paradigms. For example, the aforementioned public health risk factor analysis can be cross-purposed to posit circumstances that promote resistance to symptomatic infection or to more tangibly severe manifestations. Indeed, while much of the world is now assumed to have been exposed to SARS-CoV-2, a persistent sub-population exists that, despite not necessarily pursuing rigorous isolation, has experienced few or no symptoms of infection [33,34,35,36,37]. This resistance suggests tangible factors, likely independent of the virus itself, that may enable some individuals to avoid illness. Identifying and characterizing these features might help to inspire novel therapeutic or prophylactic paradigms, thus incentivizing resistivity/vulnerability analysis [33,34,35,36,37].

Genetic factors postulated to protect against COVID-19 infection have been reviewed elsewhere [38], with much attention being paid to aspects of host physiology that may enhance immune-driven clearance of the SARS-CoV-2 virus [38,39,40]. In particular, it is noted that some people display T-cell activity that is measurably more effective in targeting highly conserved features of the virion architecture, leading to rapid viral clearance, even without elevated antibody production, which would normally be indicative of current or prior exposure [37,41]. Other mechanistically vague trends have been noted, such as potential risk reduction among people with pre-existing food allergies [42] and possibly also prior long-term usage of endocrine-modulating drugs like Tamoxifen [43,44]. Ultimately, the breadth and scatter of possible resistance factors has hindered novel drug development, which might have coalesced around any consensus on promising host-related factors.

Given that COVID-19 infections remain difficult to treat [45], especially as new variants diverge from vaccine specificity [46], it is fortunate that the absence of consensus is not a consensus of absence. Rather, if markers from host physiology or medical history fail to support a simple relationship with COVID-19 vulnerability, the difference could implicate other influences as a basis for the relative viral resistance of a given individual. Thus, this hypothesis attains substantial novelty by distancing itself from the host–virus relationship [47] and instead explores evidence that variations in host commensal bacteria, potentially including gut, nasal, and lung microbiota, may differentially facilitate or mitigate viral entry to an extent that affects vulnerability or resistance.

While substantial research has been devoted to co-pathology among various microbes and the SARS-CoV-2 virus [48,49,50,51,52], this hypothesis aims to assess knowledge of the symbiotic host-protective capacity of some bacteria [50,53] while focusing on putative mechanisms at molecular levels of detail, with potential application to therapeutic development. In particular, a novel hypothesis is rationalized, tying key aspects of host protection to the relative capacity of different bacterial serine proteases to interact with the SARS-CoV-2 Spike protein (and other comparable viral Class I fusion proteins present in various other coronaviruses and retroviruses) in a manner that may interfere with the pathological TMPRSS2-mediated process of Spike cleavage and host-membrane permeation. 

This hypothesis has emerged from confluent evidence drawn from disparate sources, including the following:Contrast between high-risk versus low-risk segments of the human population;Differential nasal microbiotic profiles of COVID-19 patients versus non-infected people with similar demographics and experiences [54,55,56];Comparative biological assessment of the relative virion-shedding propensity of felines [57] versus various canids [57,58], augmented by microbiotic profiling performed specially for this study across a cohort of client-owned dogs;Genomic and structural biological rationalization through informatics and simulations performed herein.

## 2. Viral Host Infection: TMPRSS2-Mediated vs. TMPRSS2-Independent Mechanisms

Human TMPRSS2 is a type II trypsin-like transmembrane serine protease that has attracted substantial interest for apparent roles in facilitating host infection by various forms of influenza and coronaviruses [59,60,61,62,63]. Setting aside a persistent lack of consensus over whether TMPRSS2 is primarily expressed by endothelial or epithelial cells [64,65,66,67], it is clear that the protein is abundant on many epithelial surfaces, especially including the lung, stomach, colon, and rectum [68,69,70], but also on the surfaces of sustentacular cells in olfactory mucosa [71]. Although no firm opinion exists on the primary physiological function of TMPRSS2 [72,73], it has gained a great deal of attention for a non-physiological role in priming viral glycoproteins as a precursor to host infection. In particular, research on SARS and MERS coronavirus epidemics spurred a growing recognition of the extent to which TMPRSS2 helps to expedite host-cell fusion and clathrin-mediated endocytosis, as is shown in Figure 1A,B. While various extracellular proteases (furin, ADAM17, and others) are capable of cleaving the SARS-CoV-2 Spike protein along loop regions suitable for infectiously priming the virion, only TMPRSS2 is known to remain associated with both the viral Spike protein and the cellular ACE2 receptor after cleavage. TMPRSS2 thus accomplishes both primary cleavage of Spike and secondary integration with the host-cell membrane, achieving such steps both for coronavirus infections [61,62,74,75] and for various influenza variants [60,61,62,63].

Amid extensive efforts to design COVID-19 therapeutics, attempts have been made to disrupt TMPRSS2-related steps incumbent in SARS-CoV-2 infection [59,76,77]. Little efficacy has been demonstrated to date in such strategies, prompting speculation that TMPRSS2 may facilitate infection without being an absolute requirement for coronavirus and influenza viruses. Indeed, mechanisms for host-cell entry by more recent SARS-CoV-2 variants (especially descendants of the omicron sequence) appear to have reduced reliance on TMPRSS2 [78].

One factor that diminishes the role of TMPRSS2 in promulgating SARS-CoV-2 infections is the apparent capacity of other proteases, such as furin [79,80,81,82], ADAM-10, and ADAM-17 [83,84], to cleave the Spike protein at junctures proximal to the canonical S2′ cleavage sites favored by TMPRSS2, thus achieving comparable release of viral RNA into the host cell. An important advantage that TMPRSS2 offers the virus is a sustained association with ACE2 at the cell surface. This facilitates viral fusion directly with the host-cell plasma membrane. 

In the absence of TMPRSS2 (Figure 1C,D), a slower mode of entry may be utilized, whereby host-cell endocytosis ports the virion into an intracellular vesicle, availing an opportunity for subsequent cytosolic release by Cathepsin-B or Cathepsin-L [85,86,87]. The diminished TMPRSS2 reliance of omicron-derived variants of SARS-CoV-2 seems to be due largely to elevated exploitation of this endocytotic cathepsin infection route [78,88,89]. Reduced infectious efficiency of the cathepsin route may explain the altered symptomology exhibited by omicron variants [90,91,92] compared with the TMPRSS2-mediated cell fusion perpetrated by delta (and prior) variants [90]. In particular, the tradeoff for slower individual infection by omicron variants appears to be greater communal transmission enabled by viral preference for transmission-friendly (but TMPRSS2-poor) nasopharyngeal tissue [93,94] while posing less threat to TMPRSS2-rich tissues, such as the deep lung [95,96] and olfactory epithelium [97,98].

## 3. Role of Protease/Antiprotease Balance in Immunopathology

From a scientific perspective, the rapid, fundamental shift in pathological mechanism achieved by the SARS-CoV-2 omicron variants begs the question of “how and why”. What evolutionary pressure could have favored the rapid genetic shift [99] from an epidemiological successful delta variant to the functionally distinct omicron class, especially when the latter has seemingly offered evolutionary tradeoffs rather than unequivocal mechanistic superiority?

Some answers may tie to the elaborate reliance of various coronaviruses and influenza viruses on proteolytic function, which is regulated within host organisms by a phenomenon known as protease/antiprotease balance (PAB). PAB entails interplay within a superset of proteases, plus an adaptive array of stimulatory and inhibitory proteins and peptides [100,101,102,103]. PAB plays a major role in immunoregulatory function and helps to determine relative resistance or susceptibility to both influenza and COVID-19 infections [104,105], whereas imbalances are implicated in respiratory and gastrointestinal autoimmune conditions [106,107,108,109]. 

While the relatively static human proteome itself might not have applied evolutionary pressure to spur the complex mutations required to produce omicron, evidence that the omicron variants might have emerged through cross-species rebound (from humans to rodents and back to humans [110,111]) may embody an immunological jolt of intensity sufficient to explain the profound adaptations manifest in the omicron variants. Notably, genomic diversities in rodent proteases and antiproteases are substantially greater than the diversities in those of humans [112,113]. Furthermore, across mammalian proteomes, a disproportionate fraction of protease variations tend to occur in immune-related functions, and in comparison with rodents and some simians, several human immune-relevant protease genes are epigenetically suppressed [114]. However, species-level variations in proteolysis are also not the final determinant, since host microbiomes add substantial diversity to protease/antiprotease balance [108,115], with substantial variation in microbiome constituency being apparent across mammalian species [116,117,118,119], for reasons that include species-level traits and individual genetics, age, sex, diet, and environment.

This hypothesis offers possible insight into the questions raised at the beginning of this section, suggesting further areas of practical inquiry. Relevant questions include the following:Can we propose key players within the proteolytic machinery with major influences on SARS-CoV-2 adaptation and the determination of host vulnerability or resistance to the virus?What are the implications for future SARS-CoV-2 differentiation based on the assumption of PAB as a key evolutionary factor?Can this information be exploited for future health benefits?

## 4. Host Serpins and Proteases as Modulators of Virus Infection

Among a handful of surprising trends that have emerged regarding COVID-19 vulnerability is growing evidence that people with prior diagnoses for allergic asthma are less likely to exhibit reduced risk of severe SARS-CoV-2 infections and hospitalizations compared with the population as a whole [120,121,122,123,124]. There are various possible factors that might underlie such an observation, but it is worth noting that the most widely considered non-pathocentric hypothesis—a protective role of regular corticosteroid use by asthmatics—does not fare particularly well in view of a conflicted and generally unfavorable relationship between general corticosteroid usage and COVID-19 outcomes [125,126,127,128]. The answer, therefore, may hinge on fundamental facets in molecular physiology and in proteolytic divergence in the pathology of the two disorders.

Protease/antiprotease imbalance is a hot topic in respiratory pathology research. While COPD and bronchiectasis are more extensively studied as examples of respiratory proteolytic disorders [106,107,129,130], allergic asthma pathology is generally also considered in this category [131,132]. To relate PAB knowledge to COVID-19 pathology, we may begin by considering antiproteases, whose effects on respiratory disorders have been widely studied [104]. The most widely reported antiprotease marker candidates for allergic asthma are SERPINB10, SERPINE1, and SERPINA3. SERPINB10 appears to offer a straightforward, correlative asthmatic relationship, presenting substantial overexpression in epithelial eosinophilia for various non-infectious respiratory pathologies [133,134,135]. In terms of anti-correlators, people with SERPINA3 (also known as alpha-1 antichymotrysin/A1AC) deficiencies tend to suffer disproportionately from non-bronchial allergic asthma [136,137]. Little is known about the manner in which SERPINA3/A1AC mitigates asthmatic response, other than a general assumption that the protein has immunoregulatory and anti-inflammatory effects, perhaps through regulation of neutrophil serine proteases [138]. The respiratory epithelial presence of specific isoforms of SERPINE1 (also called plasminogen activator inhibitor-1/PAI-1) has been examined as a possible asthma correlator [139], but the full body of literature yields a complex picture, with some research suggesting that that SERPINE1/PAI-1 dysregulation is of primary interest in cases reporting very specific polymorphisms of the protein (in particular the 4G/5G form) rather than as a general factor [140,141], while other studies consign its relevance to simple prediction of adverse corticosteroid side-effects [142,143].

In terms of relevance to COVID-19 vulnerability, SERPINB10, SERPINE1, and SERPINA3 paint very different pictures. There is minimal evidence that SERPINB10 plays any differential role, whether exacerbatory or mitigative, in determining COVID-19 outcomes [144]. Patient abundance of SERPINA3/A1AC, by contrast, is proposed to be very important for COVID-19 outcomes, but the relationship is directly analogous to trends for asthmatics, such that A1AC deficiency could be considered a major risk factor for both allergic asthma and severe viral infections [145,146,147]. Finally, as a converse example, SERPINE1/PAI-1 overabundance seems to correlate strongly with both severe non-infectious respiratory pathology [141,148,149,150] and a propensity to foster coagulopathic outcomes in severe COVID-19 cases, including interrelated acute respiratory distress syndrome (ARDS) and thrombotic manifestations [151,152,153], through apparent impairment of fibrinolytic activity [151]. From this complexity, it may be reasonable to conclude that some SERPINs have significant effects on both infectious and autoimmune respiratory pathologies, but that the effects are independent of any net relationship between cases of allergic asthma and trends of diminished COVID-19 severity.

Turning the attention to prospective protease markers, one finds multiple recent studies reporting correlations between elevated levels of proteases and allergic asthmatic response [107,154]. Of particular interest is Human Airway Trypsin-like Protease (HAT; also called TMPRSS11D) having unusual abundance in samples associated with asthmatic pathology [155,156]. From the perspective of a possible role in viral infections, it is notable that although HAT/TMPRSS11D had been implicated as a possible facilitator for the original SARS-CoV pathogen [155,157], its capacity to serve in a manner analogous to TMPRSS2 in the cleavage of the SARS-CoV-2 Spike protein and subsequent host-cell induction is measurably diminished by a factor of roughly 19-fold [158]. Thus, unlike antiprotease markers shared between allergic asthma and COVID-19 analytics, HAT/TMPRSS11D represents a unique example whereby relative abundance of the protein produces an opposite effect on pathological trends in allergic asthma versus COVID-19. This fact offers a tantalizing prospect that HAT/TMPRSS11D, while behaving as an approximate analog of TMPRSS2 and other proteases in exacerbating autoimmune conditions, may exhibit functional inefficiency that kinetically interferes with the process of SARS-CoV-2 infection and replication.

## 5. Microbiotic Mimicry and Amplification of Protease/Antiprotease Dynamics

While the host PAB machinery alone provides factors that may influence trends in COVID-19 vulnerability and resistance, it is important to recall that human genetics are only one determinant of our pathophysiologically relevant proteome. A crucial component of human biochemical complexity and diversity emerges from microbial complementarity, with distinct microbiomes in upper respiratory [159,160,161,162,163], lower respiratory [115,164,165,166,167], and gastrointestinal tracts [166,167,168,169,170] all having tangibly different effects on viral outcomes. Among the various biochemical effects associated with microbial proteases are known modulations of the innate immune system including IFNγ, several interleukins, and others [171], plus proteolytic stabilization of viral capsids in a manner that may bolster virion persistence outside of hosts in moist environments [172].

A detailed survey of viral modulation by microbial proteases and antiproteases, although potentially impactful, would be a huge undertaking, far in excess of this modest hypothesis article. It is, nonetheless, interesting to identify bacterial homologs to interesting proteases, such as HAT, A1AC, and PAI-1, within the proteomes of bacterial species classes that appear to exert influence on SAR-CoV-2 infectivity. These bacterial classes include both potential exacerbators of COVID-19 (e.g., *Staphylococcus* [173] and *Pseudomonas* [174,175]) and proposed mitigators (e.g., *Dolosigranulum* [173], *Corynebacterium* [173], and *Moraxella* [52,176]). Relevant proteases and antiproteases from these species families are reported in Table 1, according to homology (% identity) and alignment coverage (% coverage) relative to HAT, A1Ac, and PAI-1.

In this table, a trivial observation is that Dolosigranulum bacteria do not seem to produce close homologs to the COVID-19-relevant SERPINs or to the TMPRSS family of serine proteases. This would suggest that if *Dolosigranulum* bacteria do help to mitigate the threats posed by SARS-CoV-2, the mechanism is not analogous to prominent host proteolytic biochemistry and may actually not involve PAB at all. Beyond this simple observation, it may be helpful to establish context for the reported numbers. In general, larger values for sequence alignment coverage are indicative of conservation of multiple functional domains, which increases the likelihood that one protein shares not only similar primary function but possibly also comparable secondary (or even tertiary) functions and may also exhibit preferences for similar subcellular milieu. This would thus imply, for example, substantial analogies between *Staphylococcus* and *Pseudomonas* homologs to human PAI-1 and an even greater analogy for the *Moraxella* homolog to SERPINa3. The fraction of aligned sequences in the various other pairings suggests effective conservation of at least one functional domain in all cases, except for the case of *Pseudomonas*, which likely has no functional homolog to human SERPINa3 (i.e., the 16% sequence coverage and no overlap with the characteristic SERPIN center inhibitory loop imply poor candidacy).

By comparison, sequence identity is a stronger guarantee of the conservation of local structural and functional attributes. While high sequence identities offer proportionally higher probability of structural and functional conservation, there is an empirically very useful lower bound of 30% identity, above which a pair of proteins may be considered homologous. This boundary, while semi-arbitrary, has withstood decades of practical application, with the assessment that roughly 90% of all protein pairs with at least 30% sequence identity across known functional domains exhibit similar three dimensional structures and comparable functions for those domains [177]. This assumption amplifies the prospective significance of the *Corynebacterium* homolog relative to human SERPINe1; the *Pseudomonas* homolog versus human SERPINa3; and the *Staphylococcus*, *Pseudomonas*, and *Moraxella* homologs to human HAT. It should also be mentioned that functional analogies frequently exist between protein pairs with sequence identities slightly less than 30% [178], thus implying that tangible homology is possible among all putative pairings in Table 1, other than SERPINa3/PNB35482.1 (*Pseudomonas*).

From weighing levels of both sequence coverage and identity, it is the opinion of the authors that the most interesting proteins listed in Table 1 are the *Staphylococcus* and *Pseudomonas* homologs to SERPINe1, the *Staphylococcus* and *Moraxella* homologs to SERPINa3, and the *Pseudomonas* and *Moraxella* homologs to HAT. All of these bacterial homologs to human SERPINs are known to have protease inhibitory function, while both of the bacterial HAT homologs are known proteases. Admittedly, the marginal sequence homologies cannot be used to definitively argue that the antiprotease specificities of bacterial SERPIN homologs closely mirror those of the asthma-relevant and COVID-19-relevant SERPINe1 and SERPINa3, or whether the bacterial proteases might hasten or hinder the SARS-CoV-2 spike mechanism of host-cell entry. Nonetheless, the levels of sequence identity and structural conservation of functionally crucial domains (i.e., the catalytic domain on proteases and the inhibitory domain on SERPINs) imply that these bacterial homologs should produce a functional shift in PAB, quite possibly in a manner that influences the host response to invading viruses.

Given the exceptional proteomic diversity present within the pan-species protease/antiprotease spectrum, the best actionable evidence might emerge from a confluence of corroborating evidence, such as exacerbatory or inhibitory roles of bacterial species that also encode proteases or antiproteases similar to COVID-19-related host proteins. As evidence of this, it should be noted that experimental mutagenetic manipulation of human SERPINb3 transformed an antiprotease with minimal capacity for TMPRSS2 inhibition into a TMPRSS2 blocker more potent than SERPINa3 [179]. Since evolutionary pressures can foster rapid bacterial mutations, it is possible that some forms of bacteria may have adapted their proteolytic capacity for the purpose of modifying human TMPRSS2 activity, as suggested in studies relating to the synergistic interplay among TMPRSS2, androgen control, and microbiome composition [180,181,182,183].

## 6. Vulnerability Trends in Comparative Biology and Human Population

Throughout the COVID-19 pandemic, valuable research was conducted on viral transmission among animal vectors, including common companion animals such as dogs and cats. Notably, although both cats and dogs proved susceptible to contracting the SARS-COV-2 pathogen and produced antibodies, feline infections often progressed to contagious levels of viral shedding, whereas dogs, although vulnerable to acquiring a viral load, rarely displayed tangible shedding and spread [54]. There are comparably stark variations in SARS-COV-2 transmission within the broader canid families, such that coyotes are found to resemble dogs (i.e., largely impervious to SARS-CoV-2 shedding and COVID-19 symptoms), while foxes exhibit SARS-CoV-2 susceptibility comparable to that of house cats [55]. The boundaries of COVID-19 vulnerabilities among these animals thus do not clearly respect either genetics (i.e., canids versus felids) or domestication status (companion animals versus closely related wild species). A more plausible delineation may emerge from dietary considerations—dogs and coyotes are omnivorous, whereas house cats and foxes are classified as carnivores (cats are strict meat eaters; foxes are also primarily carnivorous, although food insecurity may prompt their consumption of berries and other carbohydrates). This putative factor aligns with corroborative evidence of relationships between dietary diversity and human COVID-19 vulnerability, whereby plant-predominant diets with higher consumption of vegetables, whole grains, nuts, and pulses confer statistically significant protection against moderate or severe SARS-CoV-2 infections relative to a standard western diet with more red meat, simple carbohydrates, and processed foods [184,185,186,187].

Unsurprisingly, other COVID-19 trends appear to be independent of obvious dietary influences. In particular, infected children tend to be statistically less likely to produce communicable levels of virus compared with adults, as per a 2021 study that found that children fostered contagion at a rate of about 63% that of adults [188]. Interestingly, adolescents (ages 14–17) were found, statistically, to have the lowest rates of shedding [189].

While immunological arguments may be devised to independently frame each such vulnerability trend as a distinct theme, there is evidence that the vulnerabilities may, at least in part, share common origins. Namely, it is our hypothesis that vulnerability to SARS-CoV-2 is influenced by PAB shifts due to variations in the nasopharyngeal mucosal microbiome. To further specify this hypothesis, our analysis points to evidence that some species within the genus *Moraxellaceae* may express specific proteases that interfere with the infectious capacity of SARS-CoV-2, to the point where gross symptoms and shedding may be significantly affected. In terms of the age-related fluctuations in COVID-19 vulnerability, it is noted that the genetic diversity of human respiratory microbiota increases throughout childhood, reaching a peak in late adolescence or early adulthood, before declining through the adult years [190]. It may be further noted that children tend to have a substantial fraction of *Moraxella catarrhalis* in their respiratory microbiomes, while adults have greater *Streptococcal* fractions [51,191]. Separate from the simple age-dependent microbiome argument, there is a pronounced relationship between a healthy, diversified human diet and the immunological benefits of commensal bacteria [166,192,193,194]. Namely, strong evidence suggests that greater dietary diversity tends to correspond to greater diversity in host commensal bacterial populations [195,196,197,198,199]. 

Regardless of the circumstances under which microbiotic variations emerge, numerous links may be drawn between host commensal bacterial profiles and resistance to severe or symptomatic COVID-19 infections, including a simple but statistically significant correlation between COVID-19 vulnerability and pre-existing autoimmune conditions associated with microbiotic deficiencies [200]. The microbiotic argument is substantially strengthened by clinical observations of elevated abundance of nasal *Moraxella catarrhalis* in children who have resisted COVID-19 infection relative to those with symptomatic cases [52,201]. These trends are further augmented by comparative pathology arguments, such as recognition that the relative resistance of domesticated dogs to symptomatic or shedding manifestations of COVID-19 may correspond to the widely elevated prevalence of *Moraxella* species in the nasal microbiome of individual members of the *Canis lupus familiaris* species. Given the importance of nasal mucosa in both contracting and potentially treating COVID-19 (in particular as a target for active vaccine RNA antigens [202]), we quantitatively profiled the nasal microbiome of a cohort of forty client-owned domesticated dogs (see Table 2), across which *Moraxella* sp. was the only bacterial class to be universally detected. Furthermore, *Moraxella* fractions dominated the nasal microbiome in most individual dogs, leading to an overall mean abundance roughly four times greater than the second most prevalent class (*Pseudomonas aeruginosa*).

Despite the overall disparity between *Moraxella* and *Pseudomonas* abundances within dog nasal microbiomes, it is interesting to compare the host viral effects arising from the two bacterial genera. In particular, various *Pseudomonas* species seem to influence the capacity of respiratory viruses to infect airway epithelial cells, but the net effect may be opposite to that by *Moraxella*. Specifically, *P*. *aeruginosa* was observed to suppress normal airway epithelial antiviral response in a manner that exacerbates the rate and spread of viral infections [203]. Furthermore, of particular relevance to COVID-19, one notes that P. *aeruginosa* may facilitate SARS-CoV-2 entry into airway epithelial cells by amplifying local abundance of the COVID-19-promoting host protease TMPRSS2 [175,204,205,206].

In addition to this possible effect relating to TMPRSS2 function, it is worth mentioning that other studies have proposed bacterial down-regulation of ACE2 receptors as a means for diminishing the extent of viral host-cell compromise. ACE2 expression is modulated by exposure to various antigens [207,208], as well as part of the interferon response to viral [209,210] and oncological [211,212] pathogens; thus, a microbiotic capacity for modifying or obfuscating the pathogenic up-regulatory influence may affect the host’s vulnerability to viral invasion. Indeed, host levels of *P. aeruginosa* do appear to influence ACE2 abundance, with measurable impact on ACE2-related pathological mechanisms [205,213,214].

These proposed mechanisms for an apparent capacity of *Pseudomonas* bacteria to exacerbate SARS-CoV-2 infection, in concert with evidence that composition of the nasal microbiome may play a role in determining COVID-19 vulnerability in certain subsets of the human (and broader mammalian) populations, prompt the question of whether *Pseudomonas* bacteria directly or indirectly influence viral infection of airway epithelial mucosa. Such insight may provide contra-indicative insight into how commensal microbiota might protect the host. *Moraxella*, whose presence in epithelial mucosa tends to correspond with significantly diminished COVID-19 vulnerability [53,215,216], seems like a natural point of comparison, especially given the close genetic similarity between the *Pseudomonas* and *Moraxella* genera.

## 7. Prospective Biochemical Rationalizations

Among the two prospective mechanisms for comparing *Pseudomonas* and *Moraxella* influence, the ACE2 model may explain why a greater *Pseudomonas* fraction may promote SARS-CoV-2 infections [205,213,214] but does not provide a clear rationale for why *Moraxella* might have the opposite effect, as current evidence is lacking for comparable *Moraxella*-triggered ACE2 flux. TMPRSS2, on the other hand, does provide a basis to rationalize a *Moraxella* effect. Specifically, there is substantial evidence that *Moraxella* counteracts the capacity of TMPRSS2 proteins to achieve the viral priming required for cell invasion [53,216]. Unlike the *Pseudomonas*/ACE2 case, however, it seems unlikely that the capacity of *Moraxella* bacteria to modulate TMPRSS2 action arises from direct influence on expression. Rather, the most readily measurable TMPRSS2-mediated influence of commensal bacteria seems to entail suppression of the enzyme’s activity, not abundance, as evinced by changes in TMPRSS2-triggered pathways, including the immunological interferon antiviral response [175,217], and downstream androgen-mediated angiogenic effects [181,182,218,219]. This implies that any prospective TMPRSS2-related benefit arising from an elevated *Moraxella* fraction may be attributed to effects that diminish the rate of viral Spike protein cleavage relative to basal TMPRSS2 action. This further suggests a competitive dynamic, perhaps achieved through interactions between Spike and some *Moraxella* protein that mimics and interferes with a key aspect of TMPRSS2-mediated Spike cleavage or membrane induction inherent in SARS-CoV-2 infection. 

The BLAST-P analysis (Table 1) indicated the *Moraxella* Serine Protease (MSP) protein (GenBank ID MBC7754020.1; 37.82% identical to TMPRSS2, covering 56% of the sequence alignment) as a possible candidate for TMPRSS2 interference. In comparison, the analogous enzyme from *Pseudomonas* sp. (GenBank ID WP_082643865.1; PSP for short) is slightly more phylogenetically distant from TMPRSS2 (36.21% sequence conservation; 54% coverage), but we see, in Figure 2, that PSP may exhibit a tertiary structure that is more analogous to the human protein. Specifically, MSP and PSP both show substantial sequence conservation relative to TMPRSS2 within the active catalytic triad (marked by the rose-colored “C” boxes in Figure 2), but MSP has a pair of proximal sequence deletions that are not observed for PSP. 

The largest deletion (TMPRSS2 residues 353-364; marked with a green “P” in Figure 2) lies in a peripheral loop where MSP sequence diverges from TMPRSS2 and PSP. The authors have not found clear literature evidence assigning a specific function to this loop, but it is notable that one of the MSP deletion positions in the sequence alignment corresponds to a TMPRSS2 residue (V361) within a tetrad (N358-D359-L360-V361) that has been resolved in the 7MEQ crystal structure of TMPRSS2 [220] as a distinct, H-bond-stabilized β-turn, independent of any larger β-sheet tertiary structure. Definitive evidence does not exist for whether this β-turn (present in TMPRSS2 and PSP but absent in MSP) plays a key role in determining functions and behaviors of these proteins, but it should be noted that many surface β-turns serve as recognition motifs for protein–protein interactions [221], with the most prevalent class of β-turn-binding proteins being GPCRs [221,222]. Again, acknowledging that the absence of experimental confirmation of specific binding renders any etiological assertion largely speculative, it is worth noting prior reports of TMPRSS2 activating a variety of GCPRs [223,224]. Furthermore, various COVID-19 symptoms (e.g., anosmia, congestion, neurological effects, etc.) are believed to involve GPCR dysregulation [225,226,227]. The speculative conclusion one might draw from the β-turn deletion in MSP, therefore, is that TMPRSS2 and PSP might support a pathologically important GCPR interaction on the host-cell surface, while the *Moraxella* homolog, MSP, lacks this attribute.

A smaller but more intuitively impactful deletion, marked “del” in Figure 2, is in close proximity to the proteolytic active site, corresponding to TMPRSS2 residues W306 and H307. This deletion dictates a sterically smaller MSP active site (see protruding feature marked with “B1” in Figure 3B). In combination with a steric pucker (MSP residues E229 and K236 form a protruding salt bridge marked with “B2” in Figure 3B), this may alter the MSP enzymatic selectivity relative to PSP and TMPRSS2, both of which likely have broader catalytic channels. 

To simulate how altered catalytic site topology might impact possible cleavage of the SARS-CoV-2 Spike protein, we performed rigorous Gaussian accelerated molecular dynamics (GAMD) calculations (see Appendix A) to predict the relative binding free energies and enzymatic viability of TMPRSS2, MSP, and PSP toward a 27 amino acid peptide (P809-K835) spanning the S2′ cleavage site (R815-S816). From a representative set of different initial ligand approach modes, the simulations tended to preferentially converge to one of two possible bound conformations, where the first is depicted in Figure 3A,C (mode 1: quadrant A shows conformation; quadrant C shows relevant interatomic distances), while the second is shown in Figure 3B,D (mode 2: conformation and interatomic distances). These two binding modes are accommodated by a conformational shift in the catalytic serine, as can be noted by comparing quadrants Figure 3C vs. Figure 3D, as well as the Appendix A.

To assess these prospects, binding-free-energy predictions were performed (see Table 3) to evaluate hypothetical trends associated with prospective proteolytic cleavage of Spike. The collective evidence from Figure 2 and Table 3 suggests a key distinction between the binding of Spike S2′ to TMPRSS2 versus MSP. TMPRSS2 is predicted to favor mode 1, which places the target R815 carbonyl carbon in plausibly reactive proximity to catalytic serine S401 (the separation is 4.6 Å in the instantaneous structure in Figure 3A,C but averages 4.94 ± 0.04 Å over the duration of the simulation), whereas in the context of the MSP cavity, an analogous Spike conformation encounters steric hindrance, preventing a close R815 approach to the catalytic S206 serine. Conversely, mode 2 supports a theoretically viable catalytic approach for S2′ binding to MSP, but the close approach is for K184 (4.9 Å instantaneous approach in Figure 3B,D, time-averaged to 5.11 ± 0.06 Å). PSP, meanwhile, presents yet a third scenario, whereby Table 3 reflects a preference for Spike complexation via mode 1 but places K814 (not R815) in possibly reactive proximity (time-averaged separation of 4.93 ± 0.04 Å).

The binding-free-energy predictions in Table 3 add further insight into hypothetical trends associated with prospective proteolytic cleavage of Spike. Considering that TMPRSS2, MSP, and PSP all favor trypsin-like cleavage between a cationic amino acid and an adjacent polar residue, it is not surprising that those binding modes predicted to be most energetically favorable (mode 1 for TMPRSS2 and PSP; mode 2 for MSP) also correspond to those that place cationic targets in the closest proximity to the reactive serine. The magnitudes of their predicted binding energies are also instructive. Discounting the unfavorable complexes (MSP conformation 1 and PSP conformation 2), we are left with at least one conformation for each protease whose associative free energy significantly exceeds the predicted activation barrier for TMPRSS2 proteolytic cleavage, which ranges from 15.8 to 17.1 kcal/mol for a variety of cleavable peptides [228]. 

From these energy predictions, it is reasonable to infer that the exothermic energy gained from Spike binding to TMPRSS2, MSP, and PSP may, in all three cases, provide ample energetic impetus for the enzyme to overcome the reactive barrier and cleave the peptide. While the precise reaction kinetics have not been reported for proteolytic cleavage by MSP and PSP, it is worth noting that widespread conservation of protease catalytic triad structures imposes homogeneity on reactive proficiency across the broad manifold of serine proteases [229,230,231]. Known proteolysis barriers for HIV protease (15.1–17.9 kcal/mol) [232,233], for example, bear a strong resemblance to TMPRSS2 reactive barriers. Such functional similarity across otherwise rather different enzymes implies a reasonable expectation that MSP and PSP could also support cleavage reactions within this approximate range.

There is, however, one additional enzymological factor that may help to rationalize distinctions in how TMPRSS2, MSP, and PSP may comport toward a cleavable viral Spike protein—the capacity of each enzyme to release its products. For this, very stark differences can be observed in 5 ns GAMD simulations (see Appendix A for computational details) of the egress of Spike N-terminal (Figure 4A) and C-terminal (Figure 4B) fragments from our three enzymes of interest. TMPRSS2 achieves rapid release of both fragments, whereas both of the bacterial enzymes tend to persistently sustain complexes with post-cleavage Spike fragments. Interestingly, differences manifest between PSP and MSP in adhering to the N-terminal fragment that connects to the Spike ACE2-binding domain (i.e., PSP is predicted to sustain strong binding without release, while MSP is predicted to potentially loosen the complex but then permit the rebinding of the fragment in a slightly altered conformation), but both PSP and MSP hold onto the C-terminal fragment with comparable consistency. Since free cleavage of the C-terminal portion of the SARS-CoV-2 spike protein is crucial for host infection, given its role as a molecular switch in the process of syncytia formation [234,235], our preliminary analysis suggests varied TMPRSS2, MSP, and PSP proteolytic dynamics that could translate to significantly different influences on the molecular kinetics underlying viral infection. Although neither MSP nor PSP is likely to achieve the enzymatic processing of SARS-CoV-2 with the efficiency of the TMPRSS2 interaction, this inefficiency may actually be the source of a tangible host-protective effect, such that sustained complexation with the pathologically critical Spike C-terminal cleavage product may impair the capacity of SARS-CoV-2 virions to inflict infection and compromise host lymphocyte response [235].

Ultimately, the preliminary GAMD simulations were pursued for the purposes of rationalizing a prospective role of *Moraxella* and/or *Pseudomonas* in altering the PAB dynamics required for SARS-CoV-2 proteolytic virion priming. Our results suggest that the MSP (from *Moraxella*) and PSP (*Pseudomonas*) proteases may persistently bind the viral Spike protein and perhaps achieve a form of proteolytic cleavage, but with a product clearance rate substantially lower than that for human TMPRSS2. This scenario may, via inefficiency, imply a tangible host-protective effect. Based on this prediction, combined with empirical evidence arising from a diverse set of previously published observations and trends, this hypothesis hypothesizes that given sufficient expression of *Moraxella* serine protease within the host mucosal epithelia, such catalytically inefficient MSP–spike complexes may compete with the normal pathological virion invasion of host cells to an extent that impairs the capacity of the virus to infect the host at a rate that compromises lymphocyte response [235], thus rationalizing the apparent antiviral effects of *Moraxella* commensal bacteria.

## 8. Future Directions

Even when accounting for distinct pathological trends associated with different viral variants, the outcomes of COVID-19 cases vary in ways that are difficult to explain comprehensively using host attributes alone and should be further examined. Among humans, statistical divergence manifests with varying social geography, sex, and age. Racial heritage seems to play a role, but fine-grained analysis suggests that sex, age, and socioeconomic status are stronger predictors within a given racial community [236], though no single host metric confers strong predictivity. From the perspective of comparative biology, the relative vulnerabilities of companion animals, in particular cats versus dogs, suggest that dietary diversity may be an important factor—an argument that strengthens when one explores the relative vulnerabilities of other canids, such as foxes and coyotes, whereby dogs and coyotes (both omnivorous) display reduced vulnerability to symptomatic SARS-CoV-2 infection, whereas house cats and foxes (primarily carnivores) have markedly higher susceptibility, despite foxes being genetically much closer to dogs and coyotes.

Numerous superficially plausible hypotheses may be formulated to seek unification across the many different vulnerability markers, but diet and social geography (in particular as it relates to diet and exposure to pollution and pathogens) have documented effects on host microbiome profiles, which, in turn, have attracted growing attention as possible influences on COVID-19 vulnerabilities [50,52,160,161,162,165,170,173,183,190,191,192,193,200,217]. Reasonable interpretations for how such microbe–virus interplay may arise must recognize not only the observations that bacteria use proteases to modulate host physiology in ways that benefit bacterial persistence and replication [171,237,238,239] and inter-bacterial competition [239,240,241] but also the evidence that bacterial proteolytic action can tangibly modulate viral efficacy [172]. 

The diversity of host, viral, and bacterial proteases, and their associated substrate specificities could support numerous theoretical models to account for bacterial influences on COVID-19 vulnerability. To prioritize models, there is value in finding independent observations that point toward an internally consistent paradigm. For example, a combination of prior published COVID-19 vulnerability studies (on humans and other mammals, plus our own studies on dogs) suggest that the presence of *Moraxella* in host nasal microbiomes may coincide with reduced rates of severe COVID-19 infection, in concert with documented presence of a TMPRSS2-like protease in the *Moraxella* proteome. This confluence of factors motivates the primary hypothesis of this article, which is that commensal *Moraxella* may afford a degree of host protection against SARS-CoV-2 by supplying a competitive binder for the S2′ motif on the SARS-CoV-2 Spike protein. Molecular simulations predict that while TMPRSS2 should have far greater capacity to process S2′ as a cleavage substrate, *Moraxella* serine protease (MSP) may be capable of exothermically complexing with S2′ and MSP may persistently bind the resulting cleaved fragments in a manner that could delay or abrogate subsequent viral infection steps, such as syncytia formation.

As a caveat to these observations, it should be noted that a comparably homologous protein, the serine protease from *Pseudomonas* bacteria, may exhibit similarly competitive prospects for circumventing TMPRSS2 action on SARS-CoV-2 Spike proteins, even though there is little evidence for *Pseudomonas* strains exerting a net protective effect against COVID-19. It is possible that unlike *Moraxella*, *Pseudomonas* may exert multiple opposing influences, including the fact that *Pseudomonas* presence in human nasal microbiomes is far more widely considered to have pathological effects [242,243,244], and such microbial infections may perturb host immunity in a manner that incites co-infections with SARS-CoV-2 [174,245,246]. Nonetheless, the relative prospects for viral modulation by both *Moraxella* and *Pseudomonas* strains should merit careful consideration in further exploration of this hypothesis, leading to directly testable assertions, including our suggestion that co-incubation of Vero E6 cells with (a) *Moraxella* and with (b) *Pseudomonas* should lead to diminished SARS-CoV-2 infectivity relative to sterile controls, while more immuno-realistic models will show divergence, such that *Moraxella* proves to be a more effective mitigator.

## 9. Synopsis

It is finally worth reiterating a key motivation for pursuing the evaluation of such hypotheses. Despite remarkable advances in medical and pharmaceutical technologies, viruses continue to largely evade our therapeutic arsenals. Bacteria and viruses, however, have interacted with each other for hundreds of millions of years, often attaining a degree of detente that may benefit the health of host organisms. Theoretically, such viral–bacterial dynamics may support development of an in vitro environment within which therapeutically practicable commensal microbes might be trained as adjuvant allies in our battles with known or emerging viral threats. More broadly, it is reasonable to believe that identification and characterization of microbial/viral balances might help to explain how some individuals attain tangible protection against dangerous viral pathogens, thus providing a basis for acquiring molecular mechanistic insight that could inform new strategies to mitigate future pandemics. While much of these prospects remain distinctly hypothetical, however, humanity should expect to confront other deadly viruses. In order to persevere and thrive, we must continue to mine data and anomalies in order to intuit, analyze, and potentially exploit new tools for the challenges ahead.

## Figures and Tables

**Figure 1 biotech-12-00061-f001:**
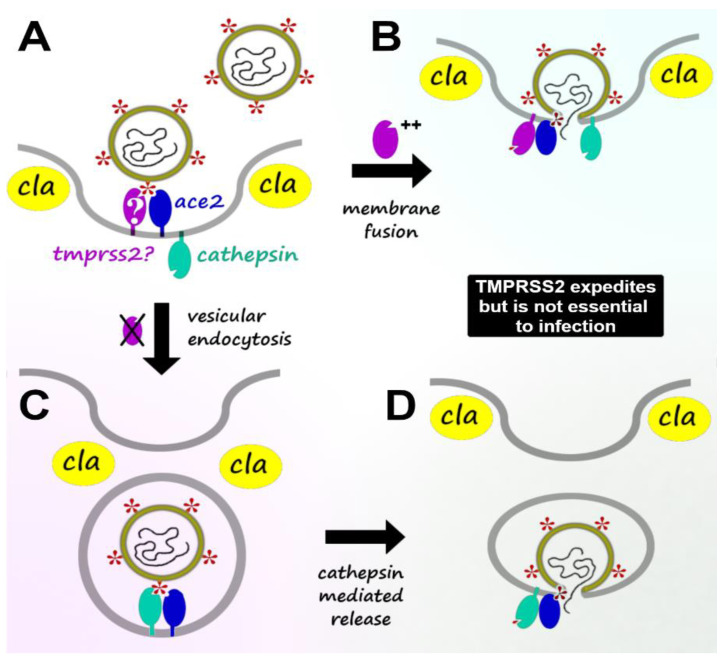
Host-cell infection, beginning with (**A**) virion binding to ACE2 receptor (blue) within a clathrin pit. (**B**) In the presence of TMPRSS2 (purple), extracellular cleavage of Spike (red) by TMPRSS2 enables membrane fusion and direct release of viral RNA into the cell. Otherwise, (**C**) clathrin (yellow cla) may mediate endocytosis, after which (**D**) various cathepsins (green) are able to release RNA from the vesicle.

**Figure 2 biotech-12-00061-f002:**
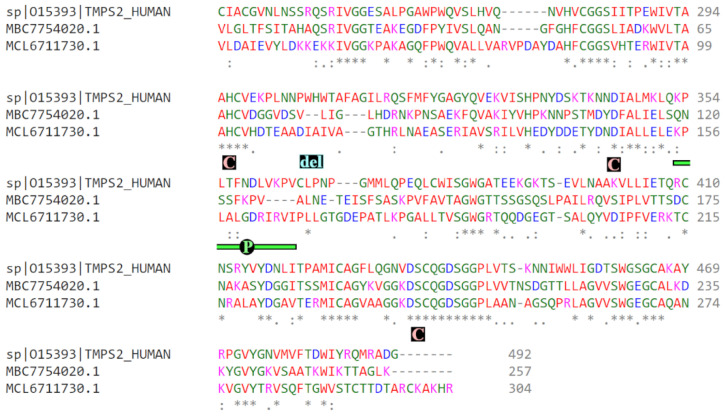
Sequence alignment for human TMPRSS2 (sp|015393), *Moraxella* serine protease (MBC7754020.1), and *Pseudomonas* serine protease (MCL6711730.1), showing catalytic triad residues (rose-colored “C”), an important channel-altering deletion (cyan-colored “del”), and a loop conferring differential protein interactions (green “P”). For sequence positions, “*” represents full conservation, “:” is partial, and “.” is weakly conserved.

**Figure 3 biotech-12-00061-f003:**
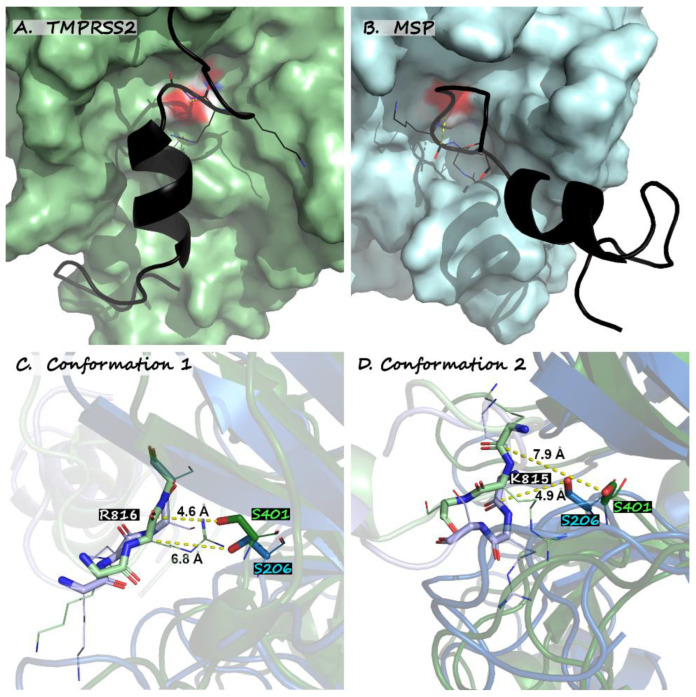
Two bound models of the SARS-CoV-2 Spike S2′ cleavable hairpin. (**A**) shows Conformation 1, bound to human TMPRSS2 (green), while (**B**) implies an alternative binding mode to the homologous *Moraxella* serine protease (blue). In (**C**), conformation 1 achieves a stable pre-reactive distance of 4.6 A relative to the TMPRSS2 catalytic serine (S441) for cleavage at R815 but does not support close approach to MSP. In (**D**), however, we see that conformation 2 supports close pre-reactive approach of cleavable K814 to MSP/S206, which is unavailable for approach to TMPRSS2/S441.

**Figure 4 biotech-12-00061-f004:**
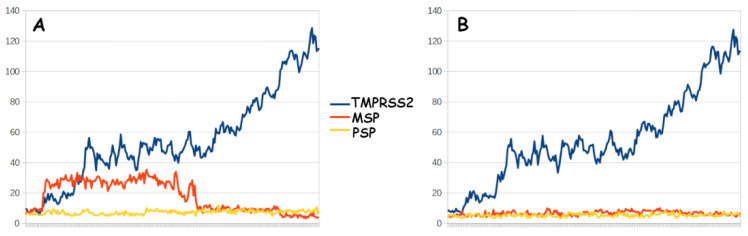
Simulated leaving group clearance for SARS-CoV-2 Spike N-terminal (**A**) and C-terminal (**B**) fragments from the proteolytic active sites of human TMPRSS2 (blue), *Moraxella* serine protease (red), and *Pseudomonas* M6 peptidase (yellow). Lines reflect the distance from proteolytic serine to the N-terminal carbonyl (**A**) or the C-terminal amine (**B**), as derived from a 5 nanosecond molecular dynamics simulation.

**Table 1 biotech-12-00061-t001:** Selected bacterial proteases and antiproteases of potential relevance to microbial modulation of SARS-CoV-2, as prioritized by sequence homology and alignment coverage relative to human proteins PAI-1, A1AC, and HAT. Microbes are listed by species family name. Microbial proteins are referenced in parentheses with GenBank accession numbers.

	PAI-1/SERPINe1		A1AC/SERPINa3		HAT/TMPRSS11D	
	% Identity	% Coverage	% Identity	% Coverage	% Identity	% Coverage
*Staphylococcus*	28.76%	90%	25.87%	87%	30.30%	31%
	(MCM1356220.1)		(MCM1356220.1)		(PAL08087.1)	
*Pseudomonas*	27.34%	93%	32.86%	16%	36.21%	54%
	(MAK72596.1)		(PNB35482.1)		(MCL6711730.1)	
*Dolosigranulum*	n/a	n/a	n/a	n/a	n/a	n/a
	(none)		(none)		(none)	
*Corynebacterium*	32.27%	60%	29.27%	57%	27.73%	51%
	(EEG25322.1)		(WP_232022389.1)		(WP_003858612.1)	
*Moraxella*	28.41%	85%	28.17%	97%	37.82%	56%
	(MBC7753780.1)		(WP_219332546.1)		(MBC7754020.1)	

**Table 2 biotech-12-00061-t002:** Abundance statistics for the five most prevalent bacterial phyla or families in nasal mucosa swabs from our microbiotic profiling of 40 client-owned companion dogs.

Phylum Family	Mean	(St. Dev.)	Count	Maximum	Minimum
*Moraxella* sp.	63.52	(25.97)	40	97.18	13.47
*Gammaproteobacteria* sp.	4.02	(10.48)	13	61.38	0.00
*Pseudomonas aeruginosa*	15.89	(22.05)	29	81.37	0.00
*Microbacteriacae* sp.	2.28	(4.45)	15	21.30	0.00
*Candidatus Gracilibacteria*	1.70	(4.69)	10	26.13	0.00

**Table 3 biotech-12-00061-t003:** Quantitative assessment of prospective modes for SARS-CoV-2 Spike S2’ binding to TMPRSS2, *Moraxella* serine protease (MSP), and *Pseudomonas* serine protease (PSP). Binding free energies from our molecular dynamics simulations are given in kcal/mol. Pre-catalytic distances correspond to mean and minimum approaches between the protease active serine and a proteolytically cleavable carbonyl, as identified in Figure 3C (for TMPRSS2) or Figure 3D (MSP).

	TMPRSS2		MSP		PSP	
	Conf. 1	Conf. 2	Conf. 1	Conf. 2	Conf. 1	Conf. 2
Binding energy	−77.45	−30.98	5.33	−19.25	−55.5	28.78
Std. err.	4.38	4.77	4.46	4.36	3.26	3.42
Catalytic approach	4.94 Å to R815 (conf. 1)		5.11 Å to K814 (conf. 2)		4.93 Å to K814 (conf. 1)	
Std. err.	0.04 Å		0.06 Å		0.04 Å	
Minimum distance	4.42 Å		4.43 Å		4.39 Å	

## Data Availability

Data are available from the authors upon request.

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
