# Peer review of "Bacterial Proteases as Potentially Exploitable Modulators of SARS-CoV-2 Infection: Logic from the Literature, Informatics, and Inspiration from the Dog"

_biotech, 2023, doi:10.3390/biotech12040061_

Round 1
Reviewer 1 Report
Comments and Suggestions for Authors
The authors of the manuscript "biotech-2527263" have addressed an important topic that is generally associated with a contribution to enhancing human health care practices. In this regard, the authors included the importance of nasal microbiome profiles in distinguishing infection rates trends among differentially affected subgroups. The topic of this paper is relevant because until now, no effective and safe drug has been found for the COVID-19 viral infection, which is mainly associated with SARS-CoV-2. Although, I have a few queries/suggestions that need to be addressed prior to consideration of publication of this study.
- The significance and novelty of findings presented in the manuscript should be further discussed and compared with previous studies in the field. It would be better if the authors can add some further comments and references described by others in order to improve the novelty and impact of this work.
- It is evident that there are many other structural and non-structural protein targets associated with the SARS-CoV-2 inhibitors are available in PDB and hence author should justify the selection of the PDB: 7WEB in this study with proper citation. Please justify the reason of choosing this protein.
- The molecular dynamics protocols should be more detailed, including the solvation model and all other simulation parameters. Additionally, information about the time step used for MD simulation and any geometric constraints applied would be valuable for validating the stability of the systems in all the analyses.
Author Response
Annotated comments are in attached file

Reviewer 2 Report
Comments and Suggestions for Authors
Lushington et al. present a very interesting hypothesis type of article regarding COVID-19. I have no reason to say no to a hypothesis paper because tons of work needs to be done before any conclusions. From my point of view, both the writing and the hypothesis itself are ok, many references are cited and that is good. So I think we can let it be published and see if any comments and further experiments would support the manuscript in the future, and vice versa.
Author Response
Annotated comments are in attached file.

Reviewer 3 Report
Comments and Suggestions for Authors
Although interesting, there are several issues with the present paper and it is also rather lengthy and hard to follow at parts. The message should be more distilled: many bacterial proteases that could potentially interact with SARS-CoV-2 to affect its severity and infectivity but more research is needed to understand the full extent of this interaction and to develop strategies to block it. The information sources should also be clearly stated to support the data put forth.
Specific comments:
1. I am not sure how "Bioinspiration from the Dog" is relevant to mention in the title since it is really less a "bioinspiration" and more evidence from animal studies. The two are not the same.
2. As per the journal's guidelines to authors, the abstract should be a total of about 200 words maximum and it should be a single paragraph without subheadings.
3. Scientific names such as "Moraxella" and "Pseudomonas" should be italicized as per convention.
4. "... – a period of prodigious scientific discovery aimed at advancing global health" - at least some supporting citations should be provided here.
5. Please state the source for Table 2. It is unclear if this is from the author's own work or a published source.
6. Authors should also mention that the nasal mucosa could also facilitate passive inhaled mRNA vaccination for SARS-CoV-2, modulating immunity (citation: ncbi.nlm.nih.gov/pmc/articles/PMC7685031).
7. What is the source for the values given in Table 3?
8. Bacterial proteases can also help SARS-CoV-2 to evade the host immune system. SARS-CoV-2 has evolved to suppress the production of IFN. Bacterial proteases can help SARS-CoV-2 to do this by cleaving proteins that are involved in IFN signaling.
9. In addition to activating SARS-CoV-2 and suppressing the host immune system, bacterial proteases can also help the virus to spread. SARS-CoV-2 can spread from person to person through respiratory droplets. However, the virus is not very stable in these droplets. Although speculative, bacterial proteases may stabilize the virus, making it more likely to spread.
10. The synopsis section itself is rather long, suggest having a separate conclusions section/paragraph.
11. "A lucky few seem “resistant” to Covid-19. Scientists want to know why (2021) STAT. https://www.statnews.com/2021/08/23/lucky- 777 few-seem-resistant-to-covid19-scientists-want-to-know-why-2" and "Haven’t had COVID yet? It could be more than just luck" - citation style inconsistent, please rectify this.
Comments on the Quality of English LanguageMinor edits advised.
Author Response

(The authors gave the same response as above.)

Round 2
Reviewer 3 Report
Comments and Suggestions for Authors
Interesting hypothesis. Thank you for the replies and revisions.
Comments on the Quality of English LanguageMinor edits only.